# Preparation Optimization of CFRP and EPDM Composite by the Co-Curing Method

**DOI:** 10.3390/ma16020503

**Published:** 2023-01-04

**Authors:** Binxiao Wei, Chen Yu, Yongping Bai, Li Liu, Jinmei He

**Affiliations:** 1MIIT Key Laboratory of Critical Materials Technology for New Energy Conversion and Storage, School of Chemistry and Chemical Engineering, Harbin Institute of Technology, Harbin 150001, China; 2Wuxi HIT New Material Research Institute Co., Ltd., Wuxi 214183, China

**Keywords:** carbon fiber reinforced polymer, epdm, co-curing, preparation optimization

## Abstract

As the requirements of aerospace technology become more rigorous, the performance of solid rocket motor (SRM) cases needs to be further optimized. In the present study, a co-curing technique was used to fabricate carbon fiber reinforced polymer (CFRP)/ethylene-propylene-diene monomer (EPDM) composites whereby the properties of CFRP/EPDM composites were adjusted by varying the temperature, heating time and type of vulcanizing agent to obtain the optimum manufacturing process. The results of crosslink density (3.459 × 10^−4^ mol/cm^3^) tested by nuclear magnetic resonance (NMR), a 90° peel strength test (2.342 N/mm), and an interlaminar shear test (ILSS = 82.08 MPa) demonstrated that the optimum mechanical properties of composites were obtained under the temperature 160 °C heated for 20 min with the curing agent DCP/S. The interfacial phase and bonding mechanism of composites were investigated by scanning electron microscopy (SEM). Thermogravimetric analysis (TGA) further indicated that EPDM/DCP/S had favorable thermal stability. This will provide valuable recommendations for the optimization of the SRM shell preparation process.

## 1. Introduction

CFRP and EPDM rubber are critical components for SRM due to their excellent performance characteristics [1,2]. CFRP has been employed as the SRM case because of its low density, high strength, dimensional stability, and corrosion resistance [3,4]. EPDM rubber has been applied as the thermal barrier for SRM because of its density (0.85 g/cm^3^), low thermal conductivity and high thermal stability [5,6,7]. However, when the rocket is launched, high-speed heat flow can generate extremely high temperatures (2000–4000 °C) and pressure (approximately 60 bar) [8,9], which can lead to the degradation and debonding of barrier [6,10]. Hence, the interfacial bonding strength between CFRP and EPDM rubber is necessary to be strong enough to resist the impact of high-speed heat flow.

The traditional method for joining EPDM rubber to CFRP is adhesive bonding [11,12]. Adhesive bonding, as compared to other bonding processes, can enable superior interface bonding strength for materials with varying polarity [13,14]. Nonetheless, a variety of surface pre-treatments are used to process the surface before materials join, which can consume a great deal of time and energy [15,16]. Another drawback of the adhesive method is that air bubbles at the interface of materials cannot be drained, resulting in stress concentrations at the interface during service, which can lead to interface failure [17,18]. To satisfy the requirements of the progressive development of SRM, the technique for bonding operations should be streamlined. Thus, a novel approach for interfacial bonding has been devised to address the issues outlined above.

Co-curing, a recently developed technology, is mainly utilized for bridging between CFRP and metal, in which the metal is adhered with CFRP by the matrix resin under the thermal and pressure load of a stamping die [16,19,20]. Shin et al. [21] studied the effects of bond parameters on tensile load-bearing capacities of co-cured steel/composite joints. Streitferdt et al. [22] designed and manufactured CFRP/steel hybrid joints by the vacuum-assisted resin infusion process. Dasilva et al. [23] discovered that metal-polymer composite joints manufactured using the co-curing process had higher mechanical behavior than secondary bonding (known as traditional adhesive bonding). Inspired by this, several scholars began to manufacture CFRP/rubber composites by the co-curing method and obtained higher interfacial bonding strength. Zhang et al. [24] made a pioneering study on the interfacial shear properties of co-cured CFRP/bromobutyl rubber composites, and the result indicated that the average shear force and shear strength were 3839.96 N and 6.4 MPa, respectively. Zhang et al. [25] systematically studied co-cured GFRP/nitrile rubber composites. The result showed that the maximum interfacial shear strength (IFSS) achieved 7.44 MPa. Zhang et al. [26] proposed a co-cured CFRP/hydrogenated nitrile butadiene rubber composite structure to improve the interfacial shear properties of traditional composites. The above research demonstrated that the maximum IFSS of specimens increased by 179.82% compared with traditional composites. So far, only a few scholars fabricated CFRP/EPDM composite using the co-curing method, which could reduce the defection and simplify the preparation process to make it cost- and energy- effective. In our previous work, we fabricated CFRP/EPDM composite with exceptional interfacial bonding properties [27]. However, the mechanical properties of CFRP and the crosslink density of EPDM rubber were significantly reduced due to high temperature over a long period of time. Hence, this research optimized the co-curing process to improve the mechanical properties of CFRP and the crosslink density of EPDM rubber by adjusting the temperature and time of the preparation process.

In this study, the crosslink density of EPDM rubber was investigated by NMR. Three different vulcanization formulations for EPDM rubber were used to study the effect of vulcanizers on the comprehensive properties of CFRP/EPDM composite. The interfacial bonding strength of the CFRP/EPDM composite was characterized by the 90° peel strength test, and the ILSS of CFRP was investigated by the interlaminar shear test. SEM was used to observe the cross-section, which explored the interfacial bonding mechanism of CFRP/EPDM composite. The process can achieve one-step integral molding of CFRP/EPDM hybrid structure for greater efficiency, which exhibits a broad application prospect in the field of SRM.

## 2. Materials and Methods

### 2.1. Materials

Carbon fibers (CFs, 6k, 1.80 g/cm^3^, 7 μm of diameter in average) were supplied by Shenzhen Guosen Pilot Technology Co., Guangdong Province, China. E-51 epoxy resin (epoxy value = 0.51, volume shrinkage: 2.5%) used as the matrix was purchased from Guangzhou Yuanzheng Chemical Co., Ltd, Guangdong Province, China. EPDM rubber (ethylene content, 49–55 wt%; 5-ethylidene-2-norbornene content, 6.7–8.7 wt%) was obtained from Jilin Longyun Chemical Co., Ltd, Jilin City, China. Dicumyl peroxide (DCP, 99%) used as the vulcanizing agent was provided by Wuhan Canos Technology Co, Ltd, Wuhan, China. Sulfur (99%), m-phenylenediamine (MPD, 99%) and triallyl-isocyanurate (TAIC, 98%) were supplied by Shanghai Aladdin Biochemical Technology CO., Ltd, Shanghai, China.

### 2.2. Preparation of EPDM Rubber

Based on the formulas shown in Table 1, EPDM rubber, DCP, and other agents were added progressively in a two-roll mill with the roll speed of 36.5 rpm under 50 °C. First, EPDM rubber was plasticized in a two-roller mill for 3 min. Thereafter, vulcanizers were added and mixed for 5 min. In addition, the EPDM blend was extruded nine times to mix all ingredients evenly after the roll gap was adjusted to less than 1 mm. To reduce residual stress, the mixture was taken from the roll and kept at ambient temperature for 24 h. The effects of S and TAIC on the properties of co-cured composites were examined in this work.

### 2.3. Preparation of Epoxy Resin Prepreg

MPD was melted into liquid under 75 °C for 10 min. Next, some epoxy resins were weighed and heated to reduce its viscosity. After that, epoxy resins and MPD were mixed uniformly. At the same time, we needed to ensure that the mass ratio between the epoxy resin and MPD is 100:15. Then, the blend was evacuated in a vacuum drying oven for 30 min to extract the air bubbles from it. Finally, the CFs were coated with the epoxy/MPD blend in an oven at 50 °C for 30 min to accomplish its wettability process and we obtained the epoxy prepreg.

### 2.4. Preparation of CFRP/EPDM Composites by the Co-Curing Method

A total of 3.5 g EPDM compounds were compressed into the mold for 24 h at 10 MPa (the height of the strip was 2 mm) to gain the EPDM strip. Following that, the EPDM strip was removed from the mold and the epoxy resin prepreg was laid flat in it. Before placing the EPDM onto epoxy prepreg, 7 cm lengths of isolation paper were set on both sides of the epoxy resin prepreg which could make the 90° peel strength test easier to carry. Eventually, the EPDM strip was deposited on the epoxy resin prepreg, and the mold was sealed firmly. The curing processes for CFRP/EPDM composites are demonstrated Table 2 with various temperatures (160 °C, 170 °C, 180 °C) and times (20 min, 30 min, 40 min, 1 h). The graphical abstract for manufacturing CFRP/EPDM composite is illustrated in Figure 1. The dimensions of the specimen were 200 mm × 5 mm × 4 mm.

### 2.5. Characterizations

The crosslink density of EPDM rubber was determined on NMR Spectroscopy (22 MHz, VTMR20-010V-I, China). NMR spectroscopy was carried on a Bruker minispec mq 20 spectrometer operated at 0.5 T with 90° pulses of 2.92 μs length and a dead time of 5.68 μs. The experiments and data analysis have been published in full detail in previous references [28,29,30].

The mechanical characterization of interfacial bonding strength was carried out by a universal testing machine (5500R, Instron, Norwood, MA, USA) to evaluate the effect of temperature, time, and type of vulcanizing agent on the interfacial bonding performance of CFRP/EPDM composite. The specimen size tested was 200 mm × 5 mm × 4 mm. For the 90° peel strength test, CFRP was fixed on a sliding rig using screws, and EPDM rubber was peeled from the substrate with a crosshead speed of 10 mm/min and an angle of 90° [31,32]. Every three specimens were determined for one group. The mean of the three values was taken as the peel strength and the error bars represent the standard error.

The CFRP/EPDM composite was fractured by means of impact. The fracture surface was smoothed with 2000 grit sandpaper so that the interface layer could be observed and then rinsed with distilled water. Field emission scanning electron microscopy spectroscopy (SEM, Hitachi S-4700, Tokyo, Japan) was performed to observe the morphology of the CFRP/EPDM composite cross-section.

The mass loss and thermal stability of EPDM rubber were investigated by a thermogravimetric analyzer (TA instrument Q500, New Castle, DE, USA). About 10 mg EPDM rubbers was heated from 25 °C to 800 °C at a heating rate of 10 °C/min under an air atmosphere.

Based on ASTM D2344 [33], a three-point short beam shear test was implemented on a universal testing machine (5500R, Instron, Norwood, MA, USA) at speed of 2 mm/min to characterize the ILSS of CFs/epoxy composite. The specimen dimensions for the interlaminar shear test were 20 mm × 5 mm × 2 mm. The test result was the average of three specimens.

## 3. Results and Discussion

### 3.1. The Crosslink Density of EPDM Rubber

The influence of temperature, time, and type of vulcanizing agent on the crosslink densities of EPDM rubbers is presented in Figure 2. As shown in Figure 2a, with a gradient increase in temperature, the crosslink density of EPDM/DCP rubber gradually decreased, while those of EPDM/DCP/S and EPDM/DCP/TAIC were kept in a relatively reasonable level. A possible explanation for this may be that the presence of curing assistants such as S and TAIC could introduce C-S or C-O bonds into this crosslink system to improve the integrity of crosslink structure [34]. In addition, in the EPDM rubbers, the C-S, C-C, and C-O bonds were broken under the prolonged high-temperature condition [35,36], resulting in a decrease in the crosslink densities of EPDM rubbers. Consequently, the optimum curing temperature for EPDM rubber was 160 °C. As the time gradient increased, the crosslink densities of EPDM rubbers showed a continuous downward trend, with the crosslink density of EPDM/DCP/S declining the fastest (in Figure 2b). The bond energies of C-S, C-C, and C-O bonds were 276 kJ/mol, 334 kJ/mol, and 364 kJ/mol [37], respectively, in which the C-S bond had the minimum bond energy. The C-S bond was more likely to be destroyed under the prolonged high-temperature condition than other bonds. The optimum curing time for EPDM rubber is 20 min. When the temperature was 160 °C and the time was 20 min, the crosslink densities of EPDM rubbers achieved the optimum values of 3.336, 3.459, and 3.622 × 10^−4^ mol/cm^3^, respectively. The crosslink densities of EPDM/DCP/S and EPDM/DCP/TAIC were significantly greater than that of EPDM/DCP, which indicates that the incorporation of S and TAIC was conducive to improving the crosslink density of the rubber.

### 3.2. The Adhesive Strength of CFRP/EPDM

As shown in Figure 3, when the curing temperature reached 180 °C, the fracture of EPDM rubbers occurred before the peeling curves reached the plateau area during the 90° peel test. The reason may be that the tensile strength of EPDM rubbers drops so sharply due to severe thermal aging that it is impossible to measure the average peel strength. From Figure 3, we can see that the peeling curves have slight fluctuations, which may derive from residual air bubbles at the interface resulting in stress concentrations during the preparation of CFRP/EPDM composites. Therefore, in the calculation of the average peel strength, these unstable data were eliminated to ensure the accuracy of the calculation. Regardless of whether the curing temperature was 160 °C or 170 °C, the maximum peel force of CFRP/EPDM composites was approximately equivalent in the range of 9–10 N in Figure 3 and the average peel strength of CFRP/EPDM composites was comparable in the range of 1.85–2.00 N/mm in Figure 4. The result illustrates that temperatures have virtually no effect on the interfacial characteristics of CFRP/EPDM composites.

As shown in Figure 5, the maximum peel force of CFRP/EPDM composites varied moderately with the curing time. The maximum peel force of CFRP/EPDM/DCP, CFRP/EPDM/DCP/S, and CFRP/EPDM/DCP/TAIC composites was around 10 N, 12 N, and 9 N, respectively. Compared to the data in Figure 4, the average peel strength of CFRP/EPDM composites in Figure 6 was significantly higher. The average peel strength of three composites reached a maximum value of 2.128, 2.342, and 2.174 N/mm, respectively, when the curing time was 20 min. This result further proves that the optimum curing temperature and time for CFRP/EPDM composites are 160 °C and 20 min. The CFRP/EPDM/S composite had the highest average peel strength. This suggests that the presence of S influences the interfacial properties of the composite. Therefore, SEM observation of cross-sections of the composites was used to further illustrate.

### 3.3. Interphase Profiles of CFRP/EPDM Composites

The microscopic appearances of CFRP/EPDM composite cross-sections are shown in Figure 7 and Figure 8. The most interesting aspect of Figure 7 and Figure 8 was that the color changed from dark to bright when going from CFRP to EPDM, and a certain width of the interfacial layer was present between bright and dark. The formation mechanism of the interfacial layer is that the viscosity of the epoxy resin will reduce with the increase of temperature before reaching gel point, resulting in its molecular chains moving easier and penetrating the EPDM rubber. In addition, the interfacial layer will be formed for the epoxy and may solidify and harden after reaching the gel point [38,39]. Interfacial entanglement and mechanical locking are the main reasons for the outstanding bond strength of composites. During the co-curing procedure, the EPDM rubber may chemically react with the substrate to form covalent bonds [40], which is helpful to gain an interfacial layer with outstanding bonding strength. As shown in Figure 7 and Figure 8, the width of the interfacial phase was irrespective of the manufacturing process and was only correlated to the type of vulcanizing agent. The widths of the interfacial phases of CFRP/EPDM/DCP, CFRP/EPDM/DCP/S, and CFRP/EPDM/DCP/TAIC were in the region of 10–20 μm, 20–30 μm, and 10–20 μm, respectively. The result may be explained by the fact that S competes with DCP to vulcanize EPDM rubber, resulting in a slow crosslinking speed of EPDM rubber with a low crosslink density and a non-dense three-dimensional network, which facilitates the penetration of epoxy resin.

### 3.4. TGA Analysis of EPDM Rubbers

Thermogravimetric (TG) and differential thermogravimetric (DTG) curves for the pyrolysis of EPDM rubbers are shown Figure 9a,b. Compared to EPDM/DCP, T_i_, T_p_, and T_f_ of EPDM/DCP/S and EPDM/DCP/TAIC are dramatically higher, but the char yield of EPDM/DCP/S and EPDM/DCP/TAIC has a slight decrease. This indicates that EPDM/DCP/S and EPDM/DCP/TAIC have higher thermal stability than EPDM/DCP [41,42]. It is attributed to the incorporation of S and TAIC, both of which could enhance the crosslink degree of EPDM rubber, thereby improving the thermal stability of EPDM rubbers [43,44]. The thermogravimetric analysis results of EPDM rubbers are described in Table 3.

### 3.5. ILSS of CFRP

As shown in Figure 10 and Figure 11, the ILSS of CFRP individually cured without EPDM and CFRP in CFRP/EPDM composites was in the range of 70 MPa–85 MPa. At a temperature of 160 °C and a heating time of 20 min, the ILSS of CFRP composites achieved optimum values of 80.42 MPa, 80.71 MPa, 82.02 MPa, and 81.11 MPa. As the heating temperature was increased and the heating time was extended, the ILSS of CFRP composites reduced. The result may be explained by the fact that high-temperature heating over long periods of time leads to the degradation of the CFRP composite [45], and the uneven heating of the CFRP composite is due to the good thermal insulation of EPDM rubber and the thermal gradient propagation in the EPDM rubber. Nevertheless, the high-temperature preparation process for a short period of time does not have a negative impact on the mechanical properties of CFRP composites, but rather has a facilitating effect, since the high temperature for a short period of time facilitates the release of residual thermal stresses within the composite, which is consistent with the previous conclusions.

## 4. Conclusions

By adjusting the temperature, heating time and curing agent of EPDM rubber, the optimum preparation of CFRP/EPDM composites has been achieved. The research results indicated that the average peel strength of the interfacial layer and interlaminar shear strength of CFRP composites was 2.342 N/mm and 82.08 MPa, respectively, under 160 °C for 20 min with DCP/S as a vulcanized system of EPDM rubber. The crosslink density of EPDM/DCP/S was 3.459 × 10^−4^ mol/cm^3^, which was higher than that of EPDM/DCP. EPDM/DCP/S had outstanding thermal stability, which could improve thermal properties of the CFRP/EPDM composite. The CFRP/EPDM composites show a distinct interfacial phase with a width of 20–30 μm observed by SEM, which may contribute to the high interfacial bond strength of the composite. The formation mechanism of the interfacial layer is that under high temperature and pressure, epoxy resin flows and diffuses into the interior of EPDM rubber, which may bond the CFRP and EPDM rubber tighter after epoxy resin reaches the gel point. This research will provide a new way for obtaining a solid rocket motor case with more cost and energy effectiveness.

## Figures and Tables

**Figure 1 materials-16-00503-f001:**
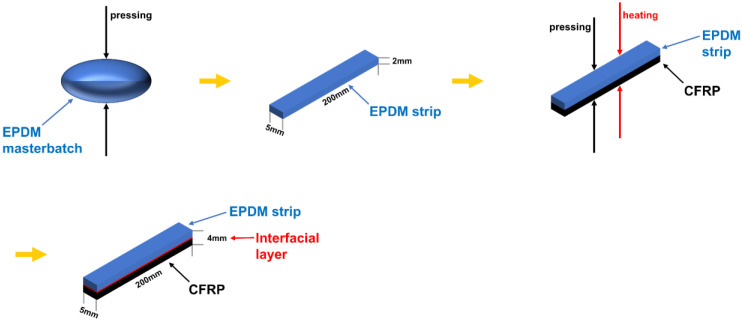
The manufacturing procedure of CFRP/EPDM composite.

**Figure 2 materials-16-00503-f002:**
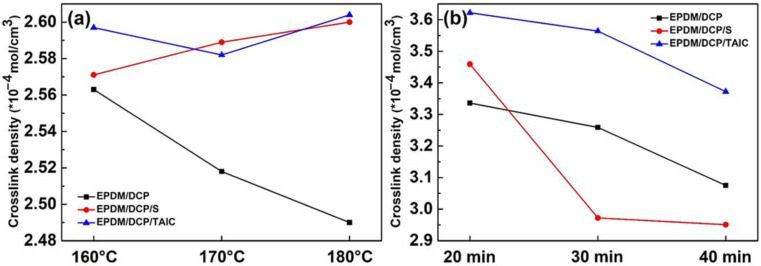
The crosslink density of EPDM rubbers (**a**) 1 h at various temperatures (160 °C, 170 °C, 180 °C); (**b**) 160 °C for various times (20 min, 30 min, 40 min).

**Figure 3 materials-16-00503-f003:**
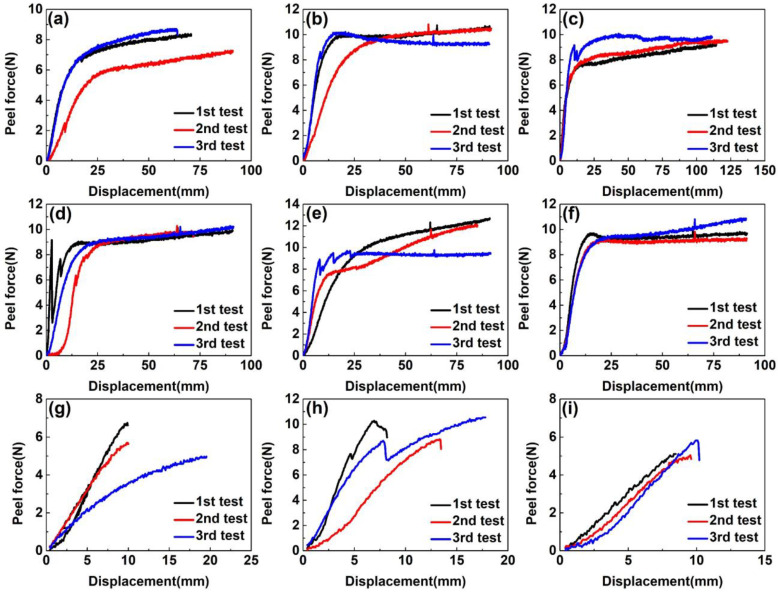
The peel curve of CFRP/EPDM composites. (**a**) CFRP/EPDM/DCP, 160 °C, 1 h. (**b**) CFRP/EPDM/DCP/S, 160 °C, 1 h. (**c**) CFRP/EPDM/DCP/TAIC, 160 °C, 1 h. (**d**) CFRP/EPDM/DCP, 170 °C, 1 h. (**e**) CFRP/EPDM/DCP/S, 170 °C, 1 h. (**f**) CFRP/EPDM/DCP/TAIC, 170 °C, 1 h. (**g**) CFRP/EPDM/DCP, 180 °C, 1 h. (**h**) CFRP/EPDM/DCP/S, 180 °C, 1 h. (**i**) CFRP/EPDM/DCP/TAIC, 180 °C, 1 h.

**Figure 4 materials-16-00503-f004:**
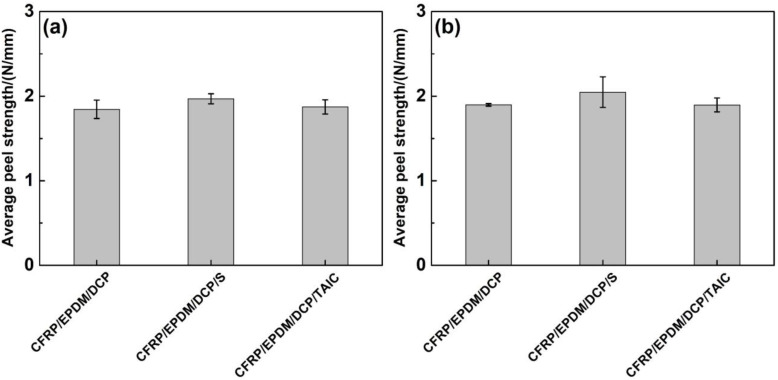
The average peel strength of CFRP/EPDM composites. (**a**) 160 °C, 1 h. (**b**) 170 °C, 1 h.

**Figure 5 materials-16-00503-f005:**
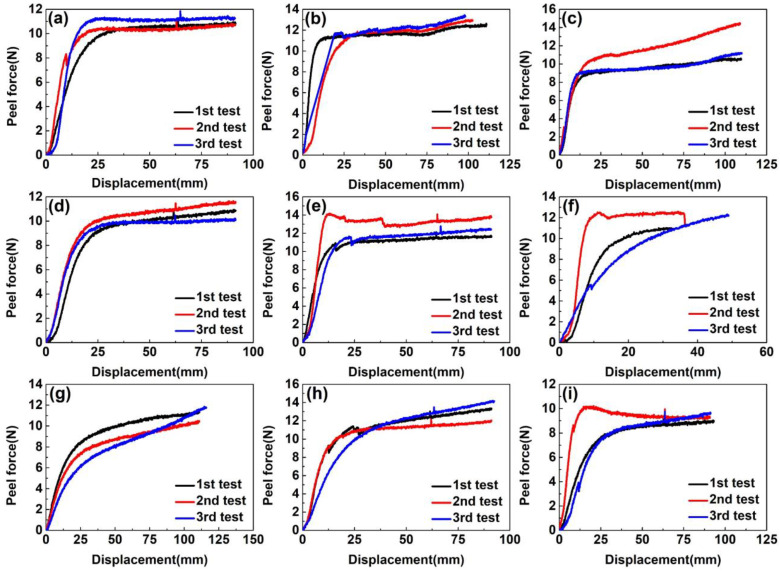
The peel curve of CFRP/EPDM composites. (**a**) CFRP/EPDM/DCP, 160 °C, 20 min. (**b**) CFRP/EPDM/DCP/S, 160 °C, 20 min. (**c**) CFRP/EPDM/DCP/TAIC, 160 °C, 20 min. (**d**) CFRP/EPDM/DCP, 160 °C, 30 min. (**e**) CFRP/EPDM/DCP/S, 160 °C, 30 min. (**f**) CFRP/EPDM/DCP/TAIC, 160 °C, 30 min. (**g**) CFRP/EPDM/DCP, 160 °C, 40 min. (**h**) CFRP/EPDM/DCP/S, 160 °C, 40 min. (**i**) CFRP/EPDM/DCP/TAIC, 160 °C, 40 min.

**Figure 6 materials-16-00503-f006:**
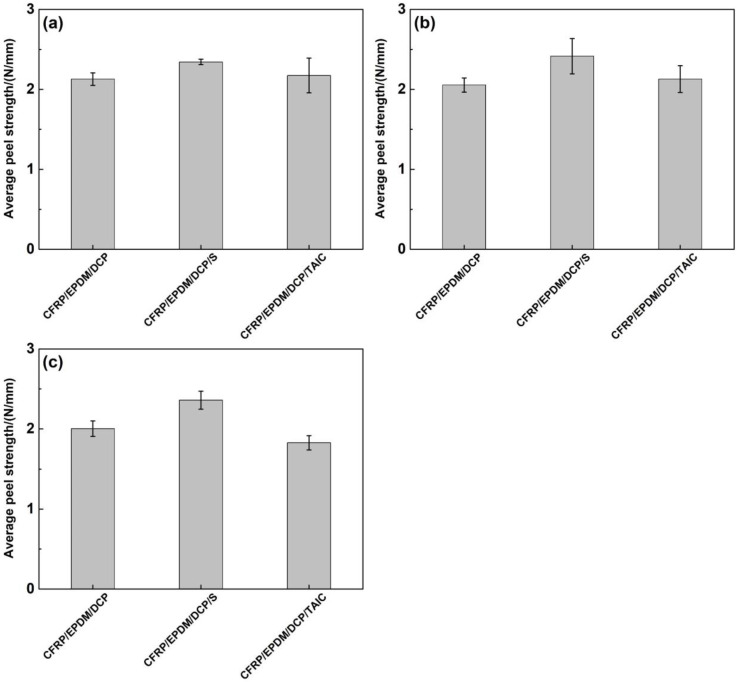
The average peel strength of CFRP/EPDM composites. (**a**) 160 °C, 20 min. (**b**) 160 °C, 30 min. (**c**) 160 °C, 40 min.

**Figure 7 materials-16-00503-f007:**
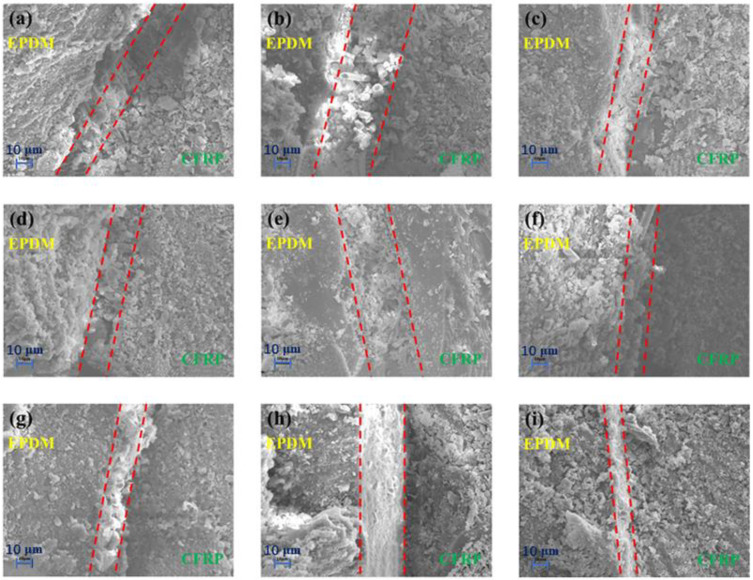
SEM images of CFRP/EPDM composite cross-sections. (**a**) CFRP/EPDM/DCP, 160 °C, 20 min. (**b**) CFRP/EPDM/DCP/S, 160 °C, 20 min. (**c**) CFRP/EPDM/DCP/TAIC, 160 °C, 20 min. (**d**) CFRP/EPDM/DCP, 160 °C, 30 min. (**e**) CFRP/EPDM/DCP/S, 160 °C, 30 min. (**f**) CFRP/EPDM/DCP/TAIC, 160 °C, 30 min. (**g**) CFRP/EPDM/DCP, 160 °C, 40 min. (**h**) CFRP/EPDM/DCP/S, 160 °C, 40 min. (**i**) CFRP/EPDM/DCP/TAIC, 160 °C, 40 min.

**Figure 8 materials-16-00503-f008:**
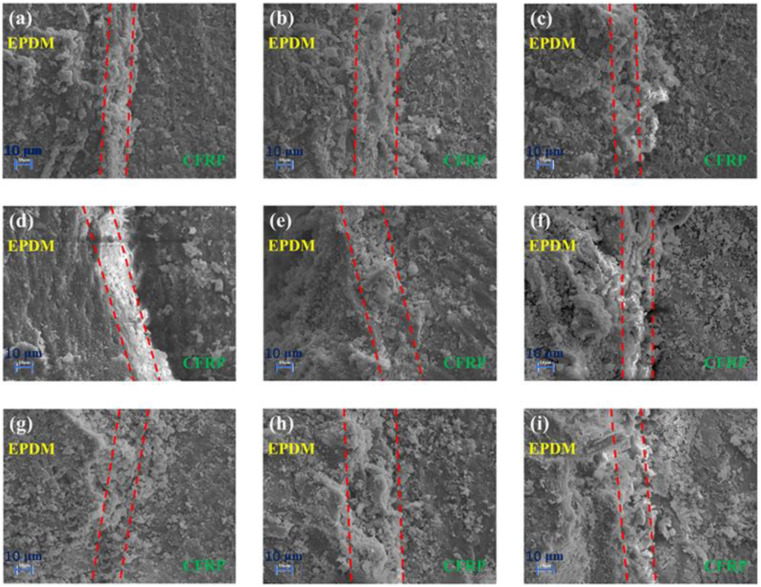
SEM images of CFRP/EPDM composite cross-sections. (**a**) CFRP/EPDM/DCP, 160 °C, 1 h. (**b**) CFRP/EPDM/DCP/S, 160 °C, 1 h. (**c**) CFRP/EPDM/DCP/TAIC, 160 °C, 1 h. (**d**) CFRP/EPDM/DCP, 170 °C, 1 h. (**e**) CFRP/EPDM/DCP/S, 170 °C, 1 h. (**f**) CFRP/EPDM/DCP/TAIC, 170 °C, 1 h. (**g**) CFRP/EPDM/DCP, 180 °C, 1 h. (**h**) CFRP/EPDM/DCP/S, 180 °C, 1 h. (**i**) CFRP/EPDM/DCP/TAIC, 180 °C, 1 h.

**Figure 9 materials-16-00503-f009:**
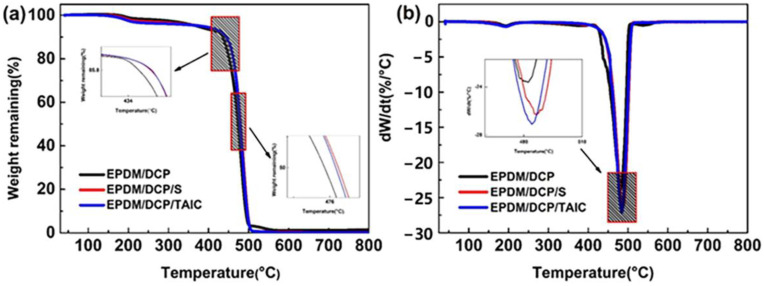
The (**a**) TGA curves and (**b**) DTG curves of EPDM rubbers.

**Figure 10 materials-16-00503-f010:**
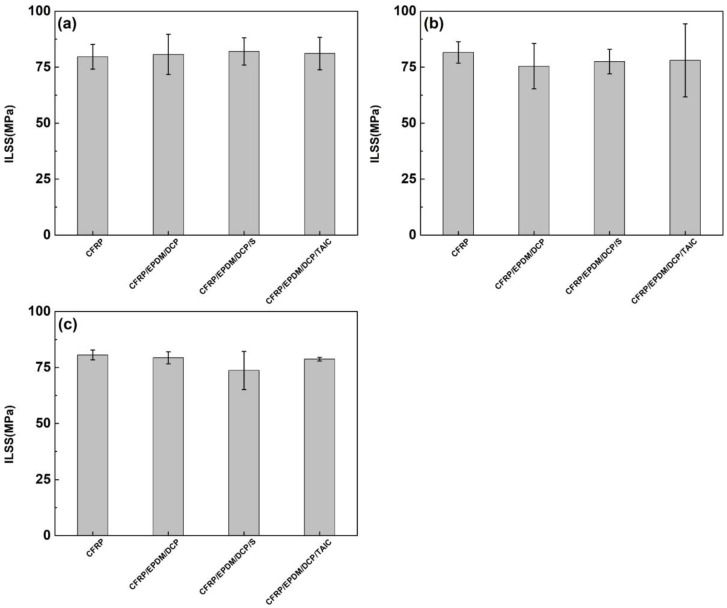
The ILSS of CFRP. (**a**) 160 °C, 20 min. (**b**) 160 °C, 30 min. (**c**) 160 °C, 40 min.

**Figure 11 materials-16-00503-f011:**
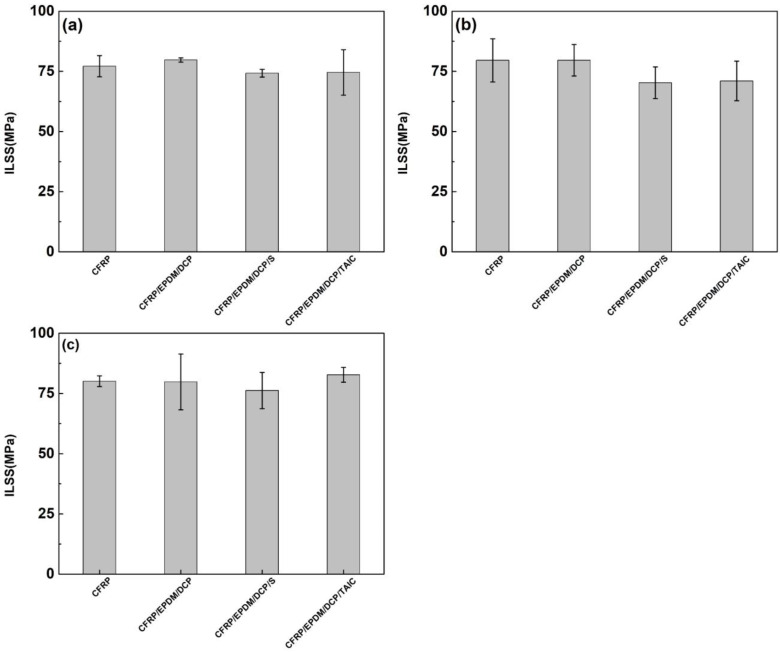
The ILSS of CFRP. (**a**) 160 °C, 1 h. (**b**) 170 °C, 1 h. (**c**) 180 °C, 1 h.

**Table 1 materials-16-00503-t001:** Formulations of EPDM rubber.

Formulation Name	EPDM (phr)	DCP (phr)	S (phr)	TAIC (phr)
EPDM-DCP	100	3.5	0	0
EPDM-DCP-S	100	3.5	0.5	0
EPDM-DCP-TAIC	100	3.5	0	5.8

**Table 2 materials-16-00503-t002:** Preparation Conditions of CFRP/EPDM composites.

Preparation Conditions	First Stage	Second Stage	Third Stage	Final Stage
1	80 °C for 2 h	120 °C for 2 h	140 °C for 1 h	160 °C for 1 h
2	80 °C for 2 h	120 °C for 2 h	140 °C for 1 h	170 °C for 1 h
3	80 °C for 2 h	120 °C for 2 h	140 °C for 1 h	180 °C for 1 h
4	80 °C for 2 h	120 °C for 2 h	140 °C for 1 h	160 °C for 20 min
5	80 °C for 2 h	120 °C for 2 h	140 °C for 1 h	160 °C for 30 min
6	80 °C for 2 h	120 °C for 2 h	140 °C for 1 h	160 °C for 40 min

**Table 3 materials-16-00503-t003:** The TGA results of EPDM rubbers.

Sample	T_i_ (°C)	T_p_ (°C)	T_f_ (°C)	Char Yield (%)
EPDM/DCP	456.83	482.21	499.63	1.379
EPDM/DCP/S	463.45	486.76	508.83	0.339
EPDM/DCPTAIC	462.31	483.8	508.83	0.123

NOTE: T_i_: initial decomposition temperature. T_p_: maximum rate of decomposition temperature. T_f_: final temperature.

## Data Availability

The data presented in this study are available in this article.

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
