# Peer review of "Preparation Optimization of CFRP and EPDM Composite by the Co-Curing Method"

_materials, 2023, doi:10.3390/ma16020503_

Round 1

Reviewer 1 Report

The article " Preparation Optimization of CFRP and EPDM Composite by the Co-curing Method "is interesting but requires correction".

Comments:

- Please complete the abstract with numerical values of the obtained results that have improved.

-Please complete the introduction in literature from recent years, e.g.

Damage assessment of CFRP laminate plate subjected to close-range blast loading: hydrocode methodology validation and case study, https://doi.org/10.1016/j.prostr.2022.01.107

Analysis of carbon fiber reinforced with resin epoxy using FEM analysis, https://doi.org/10.1016/j.matpr.2021.02.334.

- It is unclear what causes such significant differences in The crosslink density of EPDM

- Figure 7. SEM images: Please correct the scale, it is invisible.

- TGA analysis of EPDM rubbers. The section requires discussion of the results with reference to the literature.

- The peel strength results require clarification in relation to other mechanical test results

-

Author Response

December 21, 2022

Dear Diana Alexandra Minea

Editor, Materials

We thank you very much for your email dated November 28, 2022. We are pleased to submit a revised version of our manuscript entitled “Preparation Optimization of CFRP and EPDM Composite by the Co-curing Method (Materials-2093720)” for consideration to publish in Materials.

We would like to offer our most sincere thanks to our reviewers for their constructive feedback on our manuscript. We have carefully replied to all comments and followed each of the recommendations of our reviewers. We believe that all these comments and recommendations have helped us make significant enhancements to the quality of our contribution. In this rebuttal paper, you will find detailed responses to our reviewers’ comments, and a listing of all modifications made. In addition, all major changes have been highlighted in blue within the text of the revised manuscript.

We thank you and our reviewers for all the time and effort kindly invested in our contribution. We humbly hope that our careful consideration of all the comments related to the original version of our work has made the revised version that you and our reviewers will find suitable for publication in Materials.

Sincerely,

JinMei He, Ph.D

Reviewer 2 Report

The paper "Preparation Optimization of CFRP and EPDM Composite by the Co-curing Method" presents a relevant theme and within the scope of this journal, and can be considered after some corrections suggested below:

(a) Authors use many acronyms, even before their correct presentation to readers;

(b) The abstract is generally well written, however in terms of content it is generic, i.e., the authors lack an in-depth study of the quantitative results of this research;

(c) Scientific innovation is limited in the introduction of the paper, the authors must go deeper and detail what this research differs from countless others that exist on this topic, this must be evidenced together with the objectives at the end of the introduction;

(d) The state of the art of the evaluated topic needs to be improved by the authors, note that some topics are absent and need to be known with current research, such as: 10.1016/j.cscm.2022.e01407; 10.1590/1807-1929/agriambi.v24n3p187-193; 10.1016/j.cscm.2021.e00661.

(e) The methodological description, in general, is short and does not guarantee the principle of necessary reproducibility, in addition, the description of the materials is not adequate;

(f) There are numerous reference errors throughout the text, such as "Error! Reference source not found", this must be checked throughout the document;

(g) The default unit to be used should be MPa instead of "N/mm";

(h) “The formation mechanism of the interfacial layer is that the viscosity of the epoxy resin is reduced due to high temperature, resulting in its molecular chains facilitating movement and penetrating through the pores of EPDM rubber to the interior, where it begins to solidify and harden after reaching the gel point [32, 33].” The authors should better explain this excerpt;

(i) The conclusion is very short, note that several points are not effectively addressed by the authors and must be complemented.

Author Response

(The authors gave the same response as above.)

Round 2

Reviewer 1 Report

The authors make the recommended changes. Please check the spelling before publishing.

Reviewer 2 Report

ok.